# Impact of Quantitative Feedback via High-Fidelity Airway Management Training on Success Rate in Endotracheal Intubation in Undergraduate Medical Students—A Prospective Single-Center Study

**DOI:** 10.3390/jcm8091465

**Published:** 2019-09-14

**Authors:** Gunther Hempel, Wolfgang Heinke, Manuel F. Struck, Tobias Piegeler, Daisy Rotzoll

**Affiliations:** 1Department of Anesthesiology and Intensive Care, University of Leipzig Medical Center, 04103 Leipzig, Germany; 2Department of Anesthesiology and Interdisciplinary Intensive Care Medicine, District of Mittweida Hospital gGmbH, 09648 Mittweida, Germany; 3LernKlinik Leipzig—Skills and Simulation Center, University of Leipzig, 04103 Leipzig, Germany

**Keywords:** airway management, medical education, simulation, cognitive load, medical students, endotracheal intubation

## Abstract

Endotracheal intubation is still the gold standard in airway management. For medical students and young professionals, it is often difficult to train personal skills. We tested a high-fidelity simulator with an additional quantitative feedback integration to elucidate if competence acquisition for airway management is increased by using this feedback method. In the prospective trial, all participants (*n* = 299; 4th-year medical students) were randomized into two groups—One had been trained on the simulator with additional quantitative feedback (*n* = 149) and one without (*n* = 150). Three simulator measurements were considered as quality criteria—The pressure on the upper front row of teeth, the correct pressure point of the laryngoscope spatula and the correct depth for the fixation of the tube. There were a total of three measurement time points—One after initial training (with additional capture of cognitive load), one during the exam, and a final during the follow-up, approximately 20 weeks after the initial training. Regarding the three quality criteria, there was only one significant difference, with an advantage for the control group with respect to the correct pressure point of the laryngoscope spatula at the time of the follow-up (*p* = 0.011). After the training session, the cognitive load was significantly higher in the intervention group (*p* = 0.008) and increased in both groups over time. The additional quantitative feedback of the airway management trainer brings no measurable advantage in training for endotracheal intubation. Due to the increased cognitive load during the training, simple airway management task training may be more efficient for the primary acquisition of essential procedural steps.

## 1. Introduction

Endotracheal intubation is still the gold standard in airway management in anesthesiology, emergency medicine and critical care [1,2]. During daily clinical routine, for medical students and young professionals it is often difficult to practice on their individual skills in a protected environment without the potential to harm patients. Several studies suggest that it is necessary to perform at least 50–200 intubations under controlled conditions to gain confidence with the technique [3,4,5,6]. Furthermore, intubations outside the operating theater are frequently beyond controlled settings, making it more difficult to acquire competence [7]. One option to alleviate this deficiency is the use of medical simulators such as complex airway management trainers [8,9]. It is known that a “best practice-training” in a Skills Lab setting is more efficient for gaining competence than a classic “see one, do one” approach, especially when considering long-term outcomes [10]. Even videotape-based feedback can improve performance during endotracheal intubation [11]. However, one of the major flaws of available airway management trainers is their insufficient reproduction of reality. This is most notable for haptic aspects as well as for conditions of endotracheal intubation [12]. Furthermore, complications, such as injuries of the teeth or pharyngeal/tracheal mucous membranes, are often poorly represented or not even represented at all. While individual conditions vary during the intubation procedure, simulators usually remain the same. Therefore, the successful intubation of an airway management trainer does not guarantee an efficient performance in patients [13].

Apart from simulator training, structured feedback is one of the major requirements in gaining competence, especially in the field of medical education [14,15]. Feedback should be objective and direct with specific and feasible suggestions for improvement [16,17]. Thus, the combination of intensive training of a new skill and repetitive direct feedback might lead to a high cognitive effort, especially in medical students and beginners. In this context, cognitive load theory becomes relevant [18,19,20]. This theory describes the interaction between the working memory and long-term memory; whereas long-term memory can store a large amount of data, working memory as a “cache” is limited to a few pieces of information, which are remembered only for a very short time [20,21]. The evaluation of the cognitive load is individual and especially interesting at different measuring points over time during a procedure. There are various scales for measuring the cognitive load or cognitive effort; one of the most widely used is the unidimensional 9-ary rating-scale by Paas [22], which was chosen for this study.

The aim of the current study was to evaluate the potential impact of a new high-fidelity airway management trainer with quantitative and visual feedback on short- and long-term learning success. We hypothesized that the provided additional quantitative and visual simulator feedback would be able to improve the intubation skills in undergraduate medical students. Apart from that, the study tried to answer the additional research question of whether a complex simulator with additional feedback—such as the one being evaluated—might lead to a higher cognitive load when compared to classic simulator airway management training.

## 2. Materials and Methods

### 2.1. The High-Fidelity Airway Management Simulator

The “Difficult Airway Management Simulator Evaluation System” (Kyoto Kagaku Co., Ltd., Kyoto, Japan) consists of a mobile floor unit, a computer with a 16:9 touchscreen monitor including a webcam, a printer and a card reader, and the mannequin itself. The simulator contains a complex mechanical system with three engines and many different sensors, which enable the simulation of different scenarios and individual feedback (see Figure 1).

The individual simulator feedback is continuously provided to the trainee during the intubation process and again in a quantitative summary at the end of the procedure. The live feedback includes the intensity of the head extension, applied force on the teeth, as well as force and pressure application of the spatula of the laryngoscope on the mannequin’s tongue. Furthermore, the view on the vocal cords during the laryngoscopy (Cormack & Lehane classification), endotracheal tube positioning with potential one-sided lung ventilation as well as the cuff pressure are quantified (see Figure 2A). A video recording of the procedure is visualized on the screen and is available for a later review. At the end of the procedure, the feedback screen shows a summary of all the quantifiable measurements, the overall success, the duration of the procedure and the highest value of exerted pressure on the teeth (see Figure 2B).

### 2.2. Study Design

The study was designed as a single center prospective cluster-randomized trial with 4th-year undergraduate medical students at the University of Leipzig and was integrated into a 4-week problem-based learning (PBL) course focusing on emergency medicine. After approval of the institutional ethics committee (reference: 122-15-09032015), all students were informed by a handout regarding the details of the study and the relevant aspects of data collection and protection. Study participation was voluntary. The study design is shown in Figure 3.

The study commenced with a training session to learn the endotracheal training procedure (January/February, 2015). The training—as a first point of data collection (T_1_)—was held by four peer student tutors trained by two study team members (anesthesiologists) in advance. To make the training sessions comparable, all four peer student tutors were trained together based on predefined learning objectives and a handout giving a detailed course structure for the training sessions. The student tutor to student ratio was 1:5 max. Students had to enroll independently in the courses online. The different course dates were randomized in blocks to one of the two study arms: “training with additional feedback” (intervention group) and “training without additional feedback” (control group). The result of the randomization was unknown to the students up to the beginning of the course. The duration of the course was 60 min. At the beginning of the session, all students who consented to participate in the study received a questionnaire and a numeric code card to save the outcomes anonymously. The course itself started with a theoretical introduction by student peers to activate prior knowledge. The repetition of material necessary for endotracheal intubation as well as procedure indications were reviewed. Afterwards, the students were asked to record their individual cognitive load on the cognitive load rating scale for the first time during the training session (baseline—Measurement point CL_1_) [22]. In the next step, the “Difficult Airway Management Simulator Evaluation System” was introduced. The procedure of endotracheal intubation was explained step-by-step using the simulator, and typical problems and complications were discussed. Depending on whether the group was part of the intervention or the control group, the visual and quantitative feedback of the simulator’s monitor was shown to the trainee (intervention) or not (control). After this session, students had to record their individual cognitive load once again (measurement point CL_2_). The final part of the course focused on the practical training of the procedure and lasted for at least 40 min. For the practical training, the visual and quantitative feedback of the monitor was used or blinded for both the trainees and tutor. Every student had the chance to practice the procedure several times. Afterwards, the students had to record their individual cognitive load for a third and final time (measurement point CL_3_). The last training attempt was used for data evaluation. Verbal feedback and helpful hints were given by the student tutors in both groups. For the demonstration, training and later measurements of the procedure, a laryngoscope with a Macintosh-blade (size 4) was used.

The second part of the data collection was the final Objective Structured Clinical Examination (OSCE) of the PBL-course (T_2_). The OSCE exam served as the second measurement time point (T_2_) to sustain motivation in both groups and to minimize confounders (Hawthorne effect). Students were unaware of the evaluation criteria within the examination, which should have reduced the risk of misfocussing on certain special parameters from the simulator. The time between the initial training (T_1_) and the exam (T_2_) was between 1 and 4 weeks—with no significant difference between the groups due to a cluster randomization. The OSCE was held over two days in February 2015, and due to practical reasons and limitations in the number of simulators the airway-simulator was only integrated into 2 out of 4 simultaneous exam courses. These two OSCE stations were supervised by two examiners, both of whom had experience in the examination format, the simulator and the competence of endotracheal intubation itself. After a few theoretical questions, the focus of the OSCE station was the practical demonstration of the endotracheal intubation procedure (beginning with the preparation of the necessary materials and ending with the control of the correct tube positioning). All students participating in the study used their numeric code card during the exam so that the data could again be stored anonymously. During the exam, the quantitative feedback screen of the simulator was blinded to the student and the examiner to avoid interaction with the examination process.

The third and last part of the data collection, in June 2015, was the control of long-term skills acquisition approximately 20 weeks after the initial training (T_3_). The invitation to take part in this last session was spread through several local websites and forums on the internet, as well as by announcements during the regular lectures of the specific academic year. Students once again used their code cards for the anonymous data acquisition. While the intubation was performed, the quantitative feedback screen of the simulator was blinded. After completing their task, individual feedback on their performance was given by the study coordinator, integrating the quantitative feedback data obtained by the high-fidelity simulator.

### 2.3. Data Documentation and Statistical Analysis

The documentation of general aspects such as sex, allocation to the intervention or control group, prior experience in endotracheal intubation and individual cognitive load at the three measurement time points were documented on paper-based evaluation sheets. The questionnaires were analyzed with the evaluation software EvaSys v6.0 (Electric Paper Evaluationssysteme GmbH, Lüneburg, Germany). For further data analysis, SPSS (IBM SPSS Statistics v22.0 IBM Deutschland GmbH, Ehningen, Germany) was used. The data of the three different timepoints were exported to a data table in Microsoft Excel 2010 (v14.0 Microsoft Corporation, Redmond, WA, USA) and were analyzed with SPSS. A cognitive load analysis was performed via Student’s *t*-test to check for possible differences between the intervention and control group. Possible significant changes between the groups in the chronological sequence of the measurement points were tested with the Wilcoxon-test for combined samples. The analysis of differences between the groups regarding the pressure on the upper teeth was performed with the Mann–Whitney-U-test (and with the Wilcoxon-test for differences between the three time points). The analyses of the correct pressure point of the laryngoscope blade and the insertion depth of the tube between the groups were conducted with Pearson’s Chi-square test and Fisher’s exact *t*-test where appropriate—The comparison over time was performed with the McNemar test for combined samples. A *p*-value of less than 0.05 was considered to be statistically significant.

## 3. Results

312 4th year undergraduate medical students were enrolled into the study. Thirteen students did not take part in the first voluntary training course, and therefore 299 students were included in the study—114 were male (38.1%) and 185 were female (61.9%). This ratio is a comparable percentage to the overall cohort of medical students [23]. 149 students were randomized to the intervention group and obtained training with the inclusion of the additional feedback opportunities. 150 students were in the control group.

### 3.1. Analysis of Average Pressure on Upper Row of Teeth

Table 1 shows the average pressure on the upper row of teeth in both groups at the three measurement time points. In the intervention group, there was only one significant difference over time with a significant increase in the average pressure between T_2_ and T_3_ (*p* = 0.015), but with no difference when comparing T_1_ and T_3_ (*p* = 0.379). In the control group, there were no significant differences in performance during the different measurement points over time.

### 3.2. Analysis of Correct Pressure Point of the Laryngoscope Spatula on the Tongue

Table 2 shows the different percentages of the correct pressure point of the laryngoscope spatula on the tongue between the intervention group and the control group. Both the intervention group (*p* = 0.003) and the control group (*p* = 0.014) show a significantly better result at T_2_ in comparison with T_1_. Only the control group showed a persistent gain in competence until the follow-up (*p* = 0.007 for T_1_ vs. T_3_).

### 3.3. Analysis of the Correct Position of the Tube after Intubation

Table 3 shows the different percentages of the correct position of the tube after intubation between the intervention group and the control group. There was a significant deterioration for the correct endotracheal tube positioning in the intervention group (*p* = 0.021) between the first measurement point (T_1_) and the follow-up (T_3_). In the control group, there were no significant changes over time.

During the training (T_1_), there were no significant differences between the groups regarding esophageal intubation (11 in the intervention group vs. 9 in the control group (*p* = 0.632)). The same applies to the measurement during the exam (T_2_) (16 esophageal intubations in the intervention group vs. 27 in the control group (*p* = 0.180)). During the follow-up (T_3_), there were no esophageal intubations in both groups.

### 3.4. Analysis of Cognitive Load in the Training Session

The mean cognitive load at the three predefined time points during the initial training is shown for both the intervention and control group in Table 4. The cognitive load increased significantly in both groups over time (CL_1_ to CL_2_, CL_2_ to CL_3_ and CL_1_ to CL_3_; *p* < 0.001).

## 4. Discussion

### 4.1. Quality Criteria of Endotracheal Intubation

Dental injuries are one of the most frequent complications in anesthesia and the most common one during airway management procedures [24,25,26,27]. One of the main causes for dental injuries during intubation is the application of high pressure on the upper row of teeth during laryngoscopy. This risk is even higher in trainees with little experience performing the procedure, as they often use a higher force for spatula positioning [28,29]. So far, there are no available data giving information on the maximum pressure on teeth tolerated during intubation. The normal human biting force—which is between 150–200 N—could be used as an orientation [28]. In the current study, there was no difference between the two groups at any measurement time. It can therefore be concluded that quantitative feedback does not provide an additional benefit to verbal feedback, compared to verbal feedback alone. To be able to give accurate feedback when considering tooth damage, it was important that the peer student tutor was able to see the front upper teeth row during the procedure. The fact that the best results in both groups were obtained during the examination (T_2_) could be explained by an enhanced intrinsic motivation leading to better results [30]. The deterioration between the second and third measurement time points (which was significant in the intervention group) may also be explained by a decreased motivation to perform the procedure well due to missing negative consequences (i.e., mediocre exam results). Furthermore, the students had no further opportunities to obtain additional training and feedback between the examination and follow-up (T_3_). It is well established, though, that both repetitive training and feedback are essential for gaining competence in procedural skills [31].

The pressure point of the laryngoscope spatula on the tongue was a second quality criterion for endotracheal intubation. There was no significant difference between the groups for the first two measurements. In the follow-up, though, the control group had significantly better results (*p* = 0.011). This may be explained by the fact that the control group only needed to focus on the procedure during the training and subconsciously obtained a better feeling for the right positioning. In contrast, the intervention group could not concentrate on the skills performance so well and was distracted by the live feedback from the simulator monitor. The division of attention can lead to a drop in the learning success, a phenomenon which may be even more relevant in procedures performed for the first time [32]. In contrast, personal experience after appropriate instruction is demonstrably effective in the context of learning [33,34]. This is supported by the fact that, in the control group, during training the feedback possibilities were limited due to the fact that the tutor could not see the pressure point and therefore only provided information in the event of a gross malpositioning of the laryngoscope. Regarding the chronological sequence, both groups showed a significantly better performance during the exams (T_2_)—but in the follow-up, only the control group showed even better results.

With respect to the correct position of the tube at the end of the intubation, there was no significant difference between the groups. There seems to be room for improvement regarding this parameter in general, as only slightly more than 50% of the students were able to place the tube in the correct position in both groups during the exams. In contrast to the other parameters, the best results for a correct placement of the tube are captured during the training where direct feedback was possible.

### 4.2. Cognitive Load

The analysis of the cognitive load was performed with the help of the 9-ary rating scale by Paas [22]. Several other rating scales have been published in the last years [35]. However, the rating scheme by Paas has not only been extensively validated but is also easy to apply and has therefore been used in the current study. There were three time points for the measurement of the cognitive load—After the theoretical introduction (CL_1_), after the explanation and demonstration of the procedure (CL_2_) as well as after the practical training (CL_3_). The cognitive load increased significantly from one time point to the next in both groups, which indicates that the skills course increased in complexity over time. This phenomenon supports recommendations for a curriculum design with respect to cognitive load theory [36]. After the theoretical introduction (CL_1_), there was no difference in the cognitive load between the groups, which is not surprising as there was no course structure difference between both groups up to this time point. After the next measurement point, there was still no significant difference (CL_2_), but the mean values drifted apart slightly, with a slightly higher mean cognitive load in the intervention group. Finally, the difference between the groups became significant after the practical training (CL_3_), with a higher value of the mean cognitive load in the intervention group with additional audiovisual feedback (*p* = 0.008). It can therefore be concluded that additional feedback from the simulator seems to cause a higher cognitive load. This may be explained by the fact that all novices performed the procedure for the first time and had to consider simulator feedback information simultaneously. Students in the control group only had to focus on the procedure itself and obtained feedback as well as comments from the peer student tutors later, whilst students from the intervention group obtained their feedback while performing the procedure, i.e., from the screen. Perhaps this may have distracted the students from focusing on performing the procedure correctly. In conclusion, it therefore seems recommendable to start airway management training with simple airway trainers. Since the numeric value obtained on the cognitive load scale from 1 to 9 has diverse individual implications as to the stressfulness of the procedure for the student, conclusions about the possible (over)load of cognitive function for the individual student are difficult to grasp. Furthermore, the time required to perform the procedure was not recorded: it can be speculated that the cognitive load in a procedure requiring 5 min to perform may be different from a procedure taking 10 min to accomplish, even if the assessment level is the same [37]. A cognitive load of “9” is not naturally better or worse than “1”, especially under the knowledge that an appropriate range of cognitive load for learning is unknown. On the other hand, differences in individual cognitive load measurement results could explain differences in learning. For novices in particular, it is known that the high cognitive load is explainable by the fact that the attention is focused on too many (maybe superfluous) points of interest [38]. This leads to the conclusion that additional feedback, especially for novices, could rather distract from the actual task. In light of the increasing relevance of human factors for successful airway management, the implementation of cognitive aids and simulation techniques in anesthesiology training seem mandatory [39].

### 4.3. Limitations

The 6-month follow-up long-term data could be biased by a selective drop-out, but we think that the particularly good students were divided equally between both groups, so the potential risk is low. Research is needed to clarify whether these findings can also be replicated in more experienced learners (e.g., residents). Furthermore, the relevance of repeated training sessions for skills maintenance needs further investigation as well.

## 5. Conclusions

The results of the current study clearly demonstrate that there was no significant difference in the performance between the two groups regarding three quality criteria for endotracheal intubation, with the exception of the correct pressure point of the laryngoscope spatula on the tongue, which was better in the control group during the follow-up. With this exception mentioned, the additional quantitative live feedback from the high-fidelity airway management trainer leads to a significantly higher cognitive load in comparison to training with the simulator and a peer student tutor alone.

Overall, it may therefore be concluded that additional feedback from the airway management trainer—at least for students who are novices in the technique of endotracheal intubation—might bear no measurable advantage in comparison to verbal feedback from a peer student tutor only. Due to the increased cognitive load during training, it seems appropriate to start the training with simple airway trainers in order to be able to internalize the essential steps of the procedure. This is especially true in view of the high investment costs connected to a high-fidelity airway management trainer with integrated quantitative feedback.

## Figures and Tables

**Figure 1 jcm-08-01465-f001:**
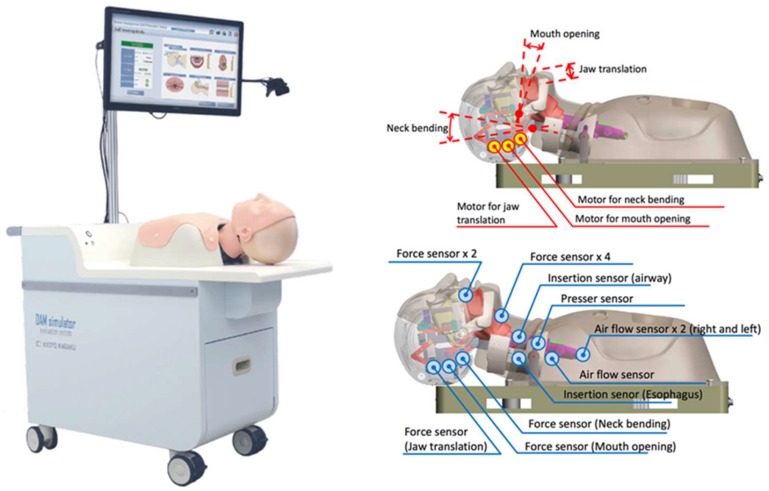
The “Difficult Airway Management Simulator Evaluation System”, including engines and sensors (Kyoto Kagaku Co., Ltd.).

**Figure 2 jcm-08-01465-f002:**
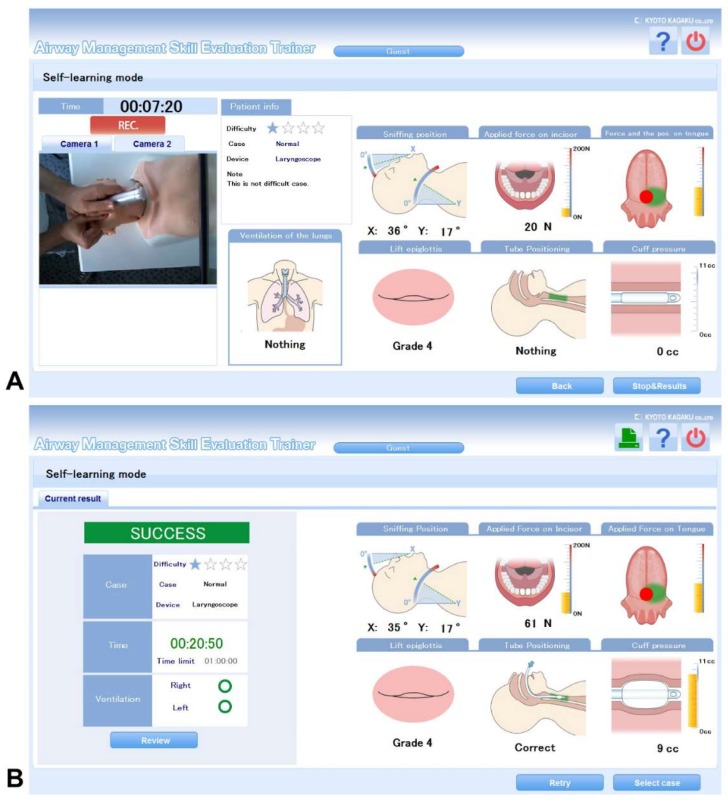
(**A**) Live feedback and (**B**) feedback summary shown by the “Difficult Airway Management Simulator Evaluation System”.

**Figure 3 jcm-08-01465-f003:**
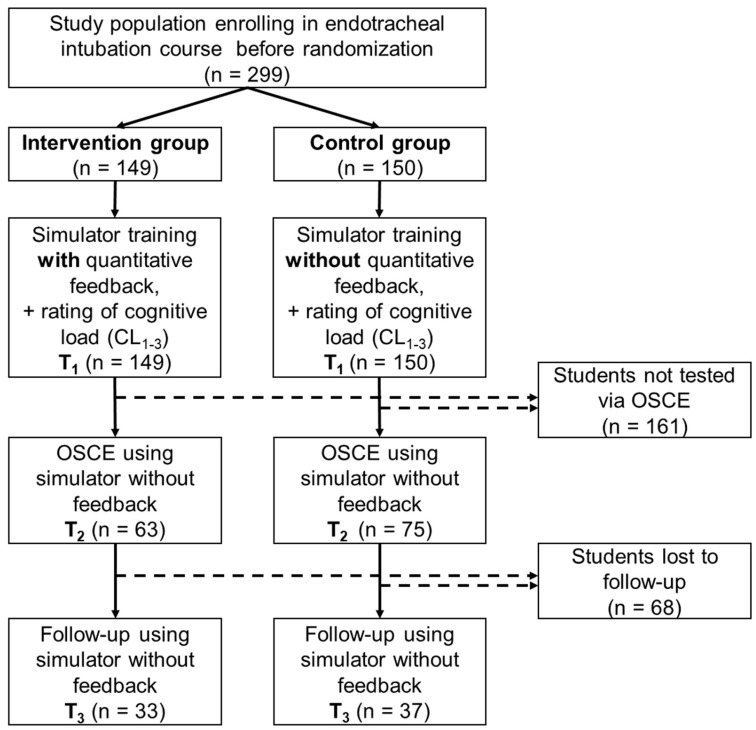
The study design involving 299 4th-year undergraduate medical students. (OSCE: Objective Structured Clinical Examination).

**Table 1 jcm-08-01465-t001:** A comparison of the average pressure on the upper row of teeth between the intervention group and control group at the three defined measurement points (T_1_ during the training session; T_2_ during the Objective Structured Clinical Examination; T_3_ during follow-up)—The pressure is described as the mean (±standard deviation) in Newton.

	*n*	Intervention Group	*n*	Control Group	Intervention vs. Control
training (T_1_)	149	57.47 (± 60.032)	150	58.09 (± 62.915)	*p* = 0.941
examination (T_2_)	63	48.35 (± 52.344)	75	47.64 (± 60.843)	*p* = 0.519
follow-up (T_3_)	33	58.85 (± 45.640)	37	51.81 (± 46.449)	*p* = 0.469

**Table 2 jcm-08-01465-t002:** A comparison of the correct pressure point of the laryngoscope spatula on the tongue between the intervention group and control group—shown in terms of the percentage of students with the correct pressure point.

	*n*	Intervention Group	*n*	Control Group	Intervention vs. Control
training (T_1_)	149	60.4% (*n* = 90)	150	52.7% (*n* = 79)	*p* = 0.177
examination (T_2_)	63	85.7% (*n* = 54)	75	76.0% (*n* = 57)	*p* = 0.152
follow up (T_3_)	33	63.6% (*n* = 21)	37	89.2% (*n* = 33)	***p* = 0.011**

**Table 3 jcm-08-01465-t003:** A comparison of the correct endotracheal tube positioning between the intervention group and control group—shown in terms of the percentage of students with the correct positioning.

	*n*	Intervention Group	*n*	Control Group	Intervention vs. Control
training (T_1_)	149	67.8% (*n* = 101)	150	58.7% (*n* = 88)	*p* = 0.102
examination (T_2_)	63	55.6% (*n* = 35)	75	54.7% (*n* = 41)	*p* = 0.917
follow up (T_3_)	33	42.4% (*n* = 14)	37	43.2% (*n* = 16)	*p* = 0.945

**Table 4 jcm-08-01465-t004:** A comparison of the cognitive load between the intervention group (*n* = 149) and control group (*n* = 150) with regard to the cognitive load at three defined points in time (CL_1_ after the theoretical introduction; CL_2_ after the practical demonstration; CL_3_ after the practical training); the cognitive load is described as the mean (± standard deviation) on a 9-step Likert scale (very, very low mental effort up to very, very high mental effort)**.**

	Intervention Group	Control Group	Intervention vs. Control
cognitive load (CL_1_)	2.79 (± 1.291)	2.85 (± 1.353)	*p* = 0.688
cognitive load (CL_2_)	3.66 (± 1.441)	3.43 (± 1.472)	*p* = 0.159
cognitive load (CL_3_)	4.75 (± 1.774)	4.19 (± 1.860)	***p* = 0.008**

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
