# Peer review of "Impact of Quantitative Feedback via High-Fidelity Airway Management Training on Success Rate in Endotracheal Intubation in Undergraduate Medical Students—A Prospective Single-Center Study"

_jcm, 2019, doi:10.3390/jcm8091465_

Round 1

Reviewer 1 Report

Paper by Hempel and colleagues covers an interesting topic, about teaching methodology for tracheal intubation, opening up new scenarios and giving a scientific demonstration for correct teaching approach.

Paper is well designed, english is fluent. 

Methods and study design, including statistics, are suitable; results are clearly presented and discussion structured and exhaustive. My only feeling is to make discussion a little bit more "user friendly" for readers who are not so familiar with the teaching issues and methodologies. As an example, I would emphasize the concept expressed in conclusions also in discussion, maybe with "simpler" approach.

ine 83: chords -> cords

line 244 "REF" a reference is probably missing.

References are updated and correct. I might suggest to include

classic Mulcaster paper on intubation learning curve

Please, aside from references 3-5, consider also 

Mulcaster JT, Mills J, Hung OR, MacQuarrie K, Law JA, Pytka S, Imrie D, Field C. Laryngoscopic intubation: learning and performance  Anesthesiology. 2003 Jan;98(1):23-7.

and a recent paper underlining the role of non-technical skills for intubation, which might have some role also in the study (Hawthorne effect?). This issue might deserve a couple of lines in discussion.

Sorbello M, Afshari A, De Hert S. Device or target? A paradigm shift in airway management: Implications for guidelines, clinical practice and teaching. Eur J Anaesthesiol. 2018 Nov;35(11):811-814

Reviewer 2 Report

The submitted paper is cluster-randomized trial of two approaches to training novices in airway management.  The authors randomly (self-segregation by blinded choice of training date) divided 299 medical students with no prior experience in airway management into two groups.  Both groups received a 60 minute didactic course (which may have been lead by peer student tutors), followed by either a hands-on practice session with an instructor (expert feedback) using a simulator providing quantitative feedback on several metrics of airway management (pressure on upper row of teeth, correct placement of laryngoscope blade, and correct placement of the endotracheal tube), or hands-on practice session with an instructor (expert feedback) using the same simulator providing no quantitative feedback (screen turned off).  The outcomes were the same quantitative metrics, as measured by the simulator, were measured at 2 time-points (a required exam several days/weeks later than; and then in a subset of participants 6 moths later).  The investigators also measured cognitive load at several time points as a potential mechanism for differences in training outcomes. The investigators found no differences in outcomes between groups at the 1st follow up period but found better laryngoscope blade placement at 6 month follow up in the control (no quantitative feedback group). The investigators have a reasonable study design, and the manuscript is well written, but there are several flaws and several changes that would improve the manuscript:

MAJOR:

Study population is very narrow. Complete novices. The results of quantitative feedback could be very different in providers at different points in the learning curve. This is not recognized in the paper The intervention is limited. One 40-minute practice session.  Again the results may have been different had the interventions been applied repeatedly. This should be recognized as a limitation of the study.  The results of this trial likely only apply to other groups performing one-time training sessions with complete novices (still a potentially large number of people) The experimental procedures are not well described.  Both groups received didactics (not clear if this was by student peers or not).  Both groups received qualitative feedback from supervisor (again not clear if student feedback or not).  The study procedures applied to both groups could be more clearly described The researchers chose, as the primary outcomes, the same variables on which the simulator provided feedback (pressure on upper row of teeth, correct placement of laryngoscope blade, and correct placement of the endotracheal tube).  The study would be improved if they reported the impact of the quantitative feedback on other important intubation outcomes. During the student examinations, what were the time to intubation, number of attempts needed, number of esophageal intubations, etc).  If the examinations were recorded the researchers may still be able to obtain this data. It is possible that the quantitative feedback forced trainees to focus on the maneuvers that would improve their quantitative score while performing poorly on other outcomes that are equally or more important The ability to measure the outcomes in only half of participants at the first testing is unfortunate but does not bias the results.  However,  selective drop-out by 6-month follow up makes the long-term data potentially biased. This data should be de-emphasized in the discussion, and this point should be noted. Cognitive load is not clearly described in the introduction or discussion. The authors do not specify if cognitive load is good or bad.  What is an appropriate range of cognitive load for learning.  Cognitive load is described as an outcome by itself, but it should be described as an explanation for differences in learning. Abstract has no description of methods or results as typical for most research papers

Minor:

What laryngoscope device was used? Page 4: monocentric should be single center" Study should be described as "cluster-randomized" Page 8, Line 213, "verbal feedback" is received by both groups. Not one or the other. Should state "quantitative feedback does not provide benefit in addition to verbal feedback, comapred to verbal feedback alone" Time between training and testing should be provided for each group
